# Effective Adsorption of Congo Red from Aqueous Solution Using Fe/Al Di-Metal Nanostructured Composite Synthesised from Fe(III) and Al(III) Recovered from Real Acid Mine Drainage

**DOI:** 10.3390/nano12050776

**Published:** 2022-02-25

**Authors:** Khathutshelo Lilith Muedi, Vhahangwele Masindi, Johannes Philippus Maree, Nils Haneklaus, Hendrik Gideon Brink

**Affiliations:** 1Department of Chemical Engineering, Faculty of Engineering, Built Environment and Information Technology, University of Pretoria, Pretoria 0028, South Africa; khathumuedi@gmail.com; 2Magalies Water, Scientific Services, Research & Development Division, Erf 3475, Stoffberg Street, Brits 0250, South Africa; masindivhahangwele@gmail.com; 3Department of Environmental Sciences, School of Agriculture and Environmental Sciences, University of South Africa (UNISA), P.O. Box 392, Florida 1710, South Africa; 4ROC Water Technologies, P.O. Box 70075, Die Wilgers, Pretoria 0041, South Africa; maree.jannie@gmail.com; 5Institute of Chemical Technology, Freiberg University of Mining and Technology, Leipziger Straße 29, 09599 Freiberg, Germany; nils-hendrik.haneklaus@extern.tu-freiberg.de; 6Td Lab Sustainable Mineral Resources, University for Continuing Education Krems, Dr.-Karl-Dorrek-Straße 30, 3500 Krems, Austria

**Keywords:** acid mine drainage, Fe/Al di-metal composite, Congo red dye, adsorption modelling

## Abstract

This study presents the first known exploration of Congo red dye (CR) adsorption by a polycationic Fe/Al Di-metal nanostructured composite (PDFe/Al) synthesised using Fe(III) and Al(III) recovered from authentic acid mine drainage (AMD). The PDFe/Al successfully removed CR from the aqueous solution. The mineralogical, microstructural, and chemical properties of the synthesised PDFe/Al adsorbent (before and after adsorption) were studied using state-of-the-art analytical instruments. The optimum conditions were observed to be 100 mg·L^−1^ CR, 1 g of the PDFe/Al in 500 mL adsorbate solution, 20 min of shaking, pH = 3–8, and a temperature of 35 °C. At optimised conditions, the PDFe/Al showed ≥99% removal efficacy for CR dye and an exceptionally high Langmuir adsorption capacity of 411 mg·g^−1^. Furthermore, a diffusion-limited adsorption mechanism was observed, with two distinct surfaces involved in the adsorption of CR from an aqueous solution. It was determined that the adsorption of CR induced internal strain and deformation within the matrices and interlayers of the PDFe/Al which resulted in a marked increase in the adsorbent pore surface area and pore volume. The remarkably high adsorption capacity could be attributed to the high surface area. A regeneration study showed that the adsorbent could be reused more than four times for the adsorption of CR. The findings from this study demonstrated the feasibility of recovering valuable minerals from toxic and hazardous AMD and demonstrated their potential for the treatment of industrial wastewaters.

## 1. Introduction

Environmental contamination by coloured, acidic, and metalliferous effluents has been an issue of topical concern globally [1,2]. This is fuelled by the magnitude of ecological and toxicological impacts posed by these wastewater streams to the receiving environment [3,4,5]. The problem of environmental contamination by coloured effluents, mainly from textile, pulp, paper, leather, and paint industries, is expected to grow primarily due to the rapid growth in population that will place proportional demand on resources to render goods and services [1]. According to the literature, about 10–25% of dye materials used in the textiles, pulp and paper, food, leather, and painting industries are lost during the dyeing or coloring process [6]. On the other hand, acid mine drainage (AMD) is rich in toxic and hazardous chemicals that need to be removed before mine water can be discharged into the environment [7]. Furthermore, the problem of mine water is expected to prevail in the coming years [8,9]. For instance, South Africa alone produces close to 360 ML of AMD per year [10], and similar or greater volumes are expected in countries with heavy mining industries such as China, Canada, and Australia [11,12]. Henceforth, mining is inevitably expected to grow in many countries since it contributes to the gross domestic products (GDPs), job creation, and socio-economic development of any given country with good mineral endowments. Consequently, mining is indispensable for both developing and developed countries. The future mining prospects project that the generation of AMD will prevail into the future [13]. Be that as it may, stringent regulatory frameworks require AMD and dyed effluents, i.e., wastewater, to be treated prior to discharge into the environment [14,15]. This will enhance the integrity of the environment and its ability to foster life. Specifically, AMD comprises elevated levels of Fe (≈3000 mg/L), Mn (≈100 mg/L), Al (≈500 mg/L) and sulphates (≈12,000 mg/L) amongst traces of other toxic and hazardous contaminants [2]. High levels of Fe and sulphates make AMD a viable source of Fe- and S-based minerals [16,17]. Various studies have demonstrated the recovery of minerals from AMD and their potential beneficiations [11,18]. Akinwekomi et al. [17] demonstrated the recovery of Fe for the synthesis of goethite, hematite, and magnetite, whilst sulphate was recovered for the synthesis of gypsum and sulphuric acid [19]. These minerals have been beneficiated for different industrial applications with Fe-based minerals receiving notable attention in water treatment. This is mainly due to their magnificent affinity to anionic species such as phosphate [20], sulphates [21], arsenic [22], and anionic dyes [23], amongst others.

Due to its stellar tenacity, chemical stability, and strong adhesive properties, Congo red dye (CR) has been widely used in printing, clothing, pulp, textile, chemical, and paint industries [24]. The primary aim of dyeing is to impart colour and enhance the aesthetic value of the material being coloured. However, product water from these industries is rich in coloured effluents since only 80% of the dye is beneficially used [25]. According to the literature, dye-rich effluents give colour to the receiving environment, hence affecting their aesthetic values and physicochemical properties [26]. Furthermore, dyes are extremely persistent in the environment, which could be attributed to their non-biodegradable (bio-recalcitrant) nature [24,26]. Amongst a wide array of discovered dyes, CR has received paramount attention [27,28]. This is mainly attributed to its alarming degree of eco-toxicity. Epidemiological studies linked CR with allergies, dermatitis, eye, skin and gastrointestinal irritation, in addition to mutation, vomiting, and diarrhoea upon exposure [29,30,31]. In addition, the unique chemical properties of CR enable it to be metabolised to benzidine in the environment. Miandad et al. [32] explicitly linked benzidine to several forms of cancer and liver tumours for living organisms after chronic exposure. In addition, the presence of CR in water prevents the penetration of sunlight in the aqua-sphere, hence affecting benthic organisms, since this shading limits the solubility of gases. Poor light penetration further compromises the process of photosynthesis in aquatic flora, with deleterious effects on both flora and fauna in aqueous environments [24,33].

To counter toxicological effects, conform to regulatory requirements, and avert potential ecological degradations, a number of technologies have been developed for the removal of CR, including adsorption [34], precipitation [35], photocatalysis [36], filtration [37], bio-sorption [38], ion-exchange [39], and integrated approaches [40]. Worryingly, some technologies fail to attenuate CR, this is attributed to the fact that CR is a very stable organic compound that is intractable with respect to chemical or biological degradation and is also very resistant to oxidising agents, aerobic digestion, heat degradation, and light degradation [31,41]. Due to low associated costs, availability, and effectiveness, adsorption has been regarded as the foremost technology for the de-colouration of water, specifically concerning CR [24,26]. For example, membrane technologies are effective but energy-intensive and require regular cleaning to regenerate the membranes. In addition, the cost associated with membrane fouling is very high, thus making the technology inaccessible for marginalised countries and communities [42]. Due to a high affinity for negatively charged chemical species, cationic metals (such as Fe(III) and Al(III)), double-layered hydroxides, and their nanocomposites have been widely employed for the removal of CR from aqueous solution, but the challenge is the use of virgin materials with the prospect of depletion in the coming years [6,23]. Ohemeng-Boahen et al. [6] reported the feasibility of using a chitosan/hematite nanocomposite hydrogel capsule fabricated via anionic surfactant gelation for the effective adsorption of CR from aqueous solution. Sriram et al. [43] successfully explored the application of Mg–Al-Layered Double Hydroxide (LDH) modified diatoms for the highly efficient removal of CR from aqueous solution. Dhal et al. [44] compared the use of ferrous oxalate, maghemite, and hematite nano-rods for the effective removal of CR dye from aqueous solution. The majority of these studies were using commercially sourced materials to synthesise the adsorbents which pose serious ecological challenges when these materials are depleted. As such, alternative ways to harvest metals such as Fe(III) and Al(III) could serve as the optimal method to preserve the environment, particularly when viewed within the circular economy context [22]. Consequently, this study seeks to recover Fe(III) and Al(III) from AMD for the synthesis of a nanostructured composite (PDFe/Al) and explore its efficacy for the removal of CR dye from aqueous solution. This is the first known study in design and execution to evaluate the feasibility of removing CR from aqueous solution using PDFe/Al synthesised from Al-Fe(III) recovered from real AMD. The findings from this study will go a long way in demonstrating the feasibility of recovering valuable minerals from toxic and hazardous mine drainage and valorise these for the effective removal of pollutants from aqueous solutions.

## 2. Materials and Methods

### 2.1. Feedstock and Sample Collection

Raw AMD was collected from a coal mine in the Mpumalanga Province of South Africa. All chemicals were purchased from Sigma-Aldrich (Darmstadt, Germany) and used as obtained without any further purification: potassium dichromate salt (K_2_Cr_2_O_7_), caustic soda (NaOH), sulphuric acid (98.5% H_2_SO_4_), and hydrochloric acid (37% HCl). CR dye stock solution was prepared using CR dye powder. Aqueous samples and solutions were prepared using ultrapure water (18.2 MΩ-cm) prepared using an Elga PURELAB^®^ Flex (ELGA LabWater, High Wycombe, UK). Experimental glassware was carefully and thoroughly cleaned before and after every use to avoid contamination.

### 2.2. Synthesis of PDFe/Al from AMD

The synthesis of PDFe/Al followed the same method as previously described [15]. To preferentially precipitate Fe(III) and Al(III) as (oxy)hydroxides, a known amount of AMD was diluted with NaOH until pH = 4.5 was measured. An overhead stirrer was used to agitate the mixture for 60 min at room temperature. The mixture was gently heated to 100 °C. with continuous stirring once the precipitation process was equilibrated. After the reaction vessel was heated and subsequently allowed to cool to room temperature, the precipitate was vacuum filtered and dried using Whatman^®^ Grade 40 ashless filter paper (Whatman PLC, Maidstone, UK). The recovered material was then crushed into a fine powder in a vibrating ball mill () at 700 rpm before being calcined at 800 °C. The samples were then sieved through a 32 μm perforated sieve () to obtain the necessary particle size, and they were stored in a plastic “zip-lock” bag to preserve them from the environment until they were used

### 2.3. Preparation of CR Dye Stock Solutions

An amount of 1 g of CR dye was dissolved in 1 L ultrapure water (18.2 M-cm) to make 1000 mg·L^−1^. The stock solution was made in a 1000 mL volumetric flask that had been filled to the specified level. For each adsorption experiment, fresh aqueous solutions were made from the stock solutions.

### 2.4. Batch Experiments

A batch-experimental approach was adopted for the removal of CR dye from aqueous solutions using the synthesised PDFe/Al. Different parameters were studied for the adsorption of CR dye, such as initial concentration, adsorbent dosage, agitation time, temperature, and initial solution pH, where 500 mL of the CR dye solution was loaded in individual volumetric flasks. For quality control and quality assurance, all experiments were conducted in triplicate and the data were reported as means. The effects of the different parameters on the adsorption process were studied as summarised in Table 1.

As depicted in Table 1, the effect of the initial CR concentration (1–200 mg·L^−1^) on the adsorption process was studied by preparing fresh aqueous solutions from concentrated 1000 mg·L^−1^ CR dye stock solutions. The effect of PDFe/Al dosage on CR dye adsorption was studied by varying different dosages of the adsorbent (0.1–5 g) and agitating them with the respective adsorbate solutions. The effect of agitation time was studied by conducting the batch adsorption experiments for CR dye adsorption for different time intervals (1–60 min). The effect of initial solution pH was studied by adjusting the pH of CR dye solutions with 0.1 M NaOH and/or 0.1 HNO_3_ to a desirable level (pH of 2–10). The effect of temperature was studied by conducting the batch adsorption experiments in a thermal shaker/incubator under different temperatures (25–65 °C). Concentrations of CR dye were analysed before and after the experiments.

To control for potential spectrophotometric interference resulting from the adsorbent matrix, the adsorbent was agitated in 500 mL distilled water overnight and, after the removal of the adsorbent, the eluent was analysed spectrophotometrically. No quantifiable change to the medium was measurable and, therefore, it was concluded that the adsorbent matrix would not cause spectrophotometric interference during the experiment.

### 2.5. Characterisation of Aqueous Samples

The pH levels of the aqueous solutions were measured using a Thermo ScientificTM Orion 3 Star portable pH meter before and after the adsorption experiments (Thermo Scientific, Waltham, MA, USA). A Mettler Toledo (Mettler Toledo^®^, Columbus, OH, USA) FiveGo EC/TDS/Salt/Temperature portable multimeter was used to measure electrical conductivity (EC), total dissolved solids (TDS), and salinity. The absorbance of the CR dye was measured using a WPA Lightwave II UV/Visible spectrophotometer (Labotec, Johannesburg, South Africa).

### 2.6. Characterisation of Solid Samples

X-ray diffraction (XRD) was used to determine the mineralogical composition and pattern of the Fe/Al and residual products, which was recorded using a Panalytical X’Pert PRO (Malvern Panalytical, Malvern, UK) equipped with a Cu-K radiation source. The Origin 2021b software program was used to deconvolute the XRD signal peak (OriginLab, Northampton, MA, USA). High-resolution scanning electron micrographs (HR-SEM-EDX) were used to examine the surface morphology and composition of solid materials (Carl Zeiss Sigma VP FE-SEM with Oxford EDX Sputtering System, Carl Zeiss AG, Oberkochen, Germany). A Perkin-Elmer Spectrum 100 Fourier Transform Infrared Spectrometer (FTIR) with a Perkin-Elmer Precisely Universal Attenuated Total Reflectance (ATR) sampler was used to examine the metal functional groups in solid samples (Perkin-Elmer^®^, Waltham, MA, USA). The synthesised PDFe/Al and subsequent residues’ Brunauer–Emmet–Teller (BET) surface area and BJH pore size were determined using BET equipment and a Micromeritics VacPrep 061 degassing system (Micromeritics Tristar II 3020, Micromeritics Instrument Corp., Ottawa, ON, Canada). The thermal stabilities of the synthesised PDFe/Al and residual products were determined using a Thermo Gravimetric Analyser (TGA) (SelectScience TGA Q500, TA instruments, Bath, UK) in ambient air at a flow rate of 50 mL·min^−1^ and a heating rate of 10 C·min^−1^.

### 2.7. Point of Zero Charge (PZC)

The Smiiklas method [20] was used to determine the point of zero charge (PZC) of the synthesised PDFe/Al. In nine separate flasks, 50 mL of 0.1 M KNO3 solution was loaded. Using 0.1 M HNO3 and/or NaOH, the pH levels of the solutions were adjusted to 2–10. After that, each flask received 0.1 g of synthesised PDFe/Al, and the mixtures were allowed to equilibrate for 24 h at 25 °C. The pH of the filtrates was then measured after the PDFe/Al suspensions were filtered.

### 2.8. Adsorption Capacity and Removal Efficiency

#### 2.8.1. Adsorption Capacity

Equation (1) was used to evaluate the adsorption capabilities of PDFe/Al for the removal of CR dye from aqueous solutions:(1)$$Q_{e}=\frac{\left(C_{0}-C_{e}\right)\cdot V}{m}$$
where *Q_e_* (mg·g^−1^) is the adsorption capacity, *C*_0_ (mg·L^−1^) is the starting concentration of CR dye, *C_e_* (mg·L^−1^) is the equilibrium concentration of CR dye, *V* (L) is the volume of the CR dye solution, and *m* (g) is the dosage of PDFe/Al, respectively.

#### 2.8.2. Removal Efficiency

Equation (2) was used to calculate the elimination effectiveness of CR dye by PDFe/Al:(2)$$\%R_{e}=\frac{\left(C_{0}-C_{e}\right)}{C_{0}}\times 100$$
where per cent *R_e_* denotes the PDFe/Al removal effectiveness, *C*_0_ denotes the initial CR dye concentration (mg·L^−1^), and *C_e_* denotes the equilibrium CR dye concentration (mg·L^−1^).

### 2.9. Desorption Study

An approach described by Kumari et al. [21] was used to investigate the regeneration of PDFe/Al. An amount of 1 g of PDFe/Al was combined with 250 mL of 150 mg·L^−1^ CR dye solution in a 250 mL Erlenmeyer flask for 90 min in a batch experiment. After equilibration, the residue was centrifuged to separate it from the supernatant. To eliminate excess CR dye ions, the recovered residual material was washed five times with 250 mL of ultrapure water and dried. After that, 250 mL of 0.1 M HNO_3_ was added to the dried sample at room temperature. The HNO_3_ extract that resulted was collected and analysed for CR dye ions. Equation (3) was used to calculate the regeneration percentage:(3)$$\%Desorption=\frac{C_{des}}{C_{0}}\times 100$$
where *C_des_* (mg·L^−1^) is the concentration of CR dye ions in the desorption eluent and *C*_0_ (mg·L^−1^) is the initial concentration of CR dye ions.

## 3. Results

### 3.1. Characterisation of PDFe/Al before and after CR Dye Adsorption

#### 3.1.1. FTIR Functional Groups

A Fourier Transform Infrared Spectrometer (FTIR) was used to characterise the functional groups of the synthesised PDFe/Al before and after CR dye adsorption, as shown in Figure 1.

As depicted in Figure 1, the transmittance bands of the synthesised PDFe/Al are shown before and after CR dye adsorption, as well as the FTIR spectrum of pure CR dye. For the raw and CR-adsorbed PDFe/Al, the high stretching of the –OH group is observed between 4000 and 3500 cm^−1^, while at 1668 cm^−1^, HOH stretching is observed [45]. Further on, as an attestation of CR dye adsorption, an increase in the sharp peak at 1134 cm^−1^, likely an indication of S=O- stretching, is observed. S=O- was also observed in the spectrum for the pure CR at 1278, 1177, and 1041 cm^–1^ [46]. A small waveband is observed at 866 cm^−1^, which could be an indication of –OH stretching in α-FeOOH [47]. Furthermore, the stretching of the Fe–O band is observed at 789 cm^−1^ The wavebands at 3740 and 2966 cm^−1^ could also indicate the stretching of the –OH group. The AlOOH band is observed to have stretched at 630 cm^−1^ [48].

#### 3.1.2. XRD Mineralogical Composition

X-ray diffraction (XRD) was used to characterise the mineralogical composition of the synthesised PDFe/Al before and after CR dye adsorption, as shown in Figure 2.

As illustrated in Figure 2, the XRD spectra of the PDFe/Al composite before and after CR dye adsorption are shown. It was observed that the XRD spectra depict the characteristic peaks for goethite—see Figure 2c—before and after adsorption, positively identifying the crystalline stricture as goethite [22]. The material also shows additional amorphous peaks after CR dye adsorption. Crystalline structures are observed to have formed at the surface of the material. The diffraction peaks between 2θ = 20° and 2θ = 40° show that the adsorption of CR dye occurred as observed from the peak size when compared to raw PDFe/Al.

From the relative positions of the fitted Gaussian peaks—shown in Figure 2a,b—as shown in Figure 2e, it can be concluded that a constant shift of −0.451° for the Bragg peaks after CR adsorption was observed. This is consistent with the significant adsorption strain experienced by the adsorbent during the adsorption of CR. According to Dolino et al. [49], the lattice mismatch parameter (Δa/a) in the direction perpendicular to the sample surface is directly proportional to the shift of the Bragg peak angle (Δθ), according to the equation Δa/a = Δθcotθ. This equation shows that a shift in the Bragg peak would result in a proportional shift in the lattice mismatch parameter, which indicates strain within the crystal lattice [49,50]. The strains within the pore structures are likely a result of competition between expansion due to decreased surface energy and contraction due to dispersion forces. These adsorption-induced strains result in the deformation of the pore structure of the adsorbent—mainly in the less rigid parts of the structure [49,50,51].

#### 3.1.3. SEM Morphology

Scanning electron microscopy (SEM) was used to characterise the morphology of the synthesised PDFe/Al before and after CR adsorption, as shown in Figure 3.

As illustrated in Figure 3, the morphological properties of the synthesised PDFe/Al composite before and after CR dye adsorption is shown. Figure 3A–C illustrate the morphology of raw PDFe/Al composites at different sizes, where non-uniform pressed-like structures of irregular agglomerates are unevenly distributed. Figure 3D–F illustrates the morphology of the PDFe/Al composite after CR dye adsorption at different sizes, where protruding plateau-like structures and small spherical structures after contacting CR-rich water can be observed.

#### 3.1.4. EDX Elemental Mapping

An energy dispersive X-ray (EDX) was used to characterise the mapping of the elemental composition of the synthesised PDFe/Al before and after CR adsorption, as shown in Figure 4.

As illustrated in Figure 4, the elemental mapping of PDFe/Al dimetal composite before and after CR dye adsorption is shown. Figure 4a–e shows that the raw PDFe/Al dimetal composite exhibited the presence of both Fe and Al, hence confirmation of co-precipitation from AMD. The presence of S and Ca are a result of the presence of these minerals in the raw AMD. Figure 4f–j shows the elemental distribution of PDFe/Al dimetal composite after CR dye adsorption. Figure 4k was generated by comparing the respective pictures in Figure 4a–j with a pure black picture, using the method described previously [22]. The quantitative analyses in Figure 4k show that no significant change in the surface coverage for the different species before and after CR adsorption was discernable—this was confirmed by Student’s t-test at the 5% significance level with t-statistics in excess of 35% for all surface species indicating that no significant difference between the surface coverage for the different species could be observed.

#### 3.1.5. TGA Thermal Stability

Thermo-gravimetric analysis (TGA) was used to determine the thermal stability of the synthesised PDFe/Al before and after CR adsorption, as shown in Figure 5.

As illustrated in Figure 5, the thermal stability of PDFe/Al before and after CR dye adsorption is shown. The graphical results show that the calcination process occurred in three phases, where the first stage involves the loss of moisture from the material between 100 and 350 °C. The second stage involves the loss of HOH that is chemically bound to the material at 400–550 °C, while the third stage involves the loss of the hydroxyl group (–OH) at temperature >550 °C [22].

#### 3.1.6. BET Surface Area and Porosity

BET surface area analysis was used to determine the surface area and porosity properties of PDFe/Al before and after CR dye adsorption, as illustrated in Figure 6 and Table 2.

As illustrated in Table 2, BET analysis was used to characterise the surface area and porosity properties of the PDFe/Al composite. It was observed that the adsorbent showed a massive increase in the surface area of approximately 37 m^2^.g^−1^ to 134 m^2^.g^−1^. Further analysis of the pore volume (Figure 6a) and area (Figure 6b) distributions show large spikes in both volume and area in the range pore diameters <20 nm when comparing the raw PDFe/Al composite to the CR-adsorbed material. In addition, the hysteresis loops for both raw and CR-adsorbed PDFe/Al (Figure 6c) show H4-type hysteresis behaviour, which is associated with slit-like pores [52]. Li et al. [53] reported that C, H, O, Na, Al, and Fe are atoms that can play a significant role in strengthening the Van der Waals forces of the material. The adsorption of CR dye, which contains C, H, O, and Na, causes significant changes in the internal forces within the material. These results are consistent with Figure 2e, which shows a significant adsorption-related strain within the adsorbent, resulting in the significant deformation of the materials [51]. Consequently, nanosized cracks form within the materials, which increases the internal pore volume, the micropore volume, and the micropore area (see Table 2). These cracks are then exhibited as slit-like pores, giving the characteristic H4-type hysteresis loop (Figure 6c) [52], which results in a corresponding increase in the adsorption capacity of the material due to an increased surface area available for adsorption [27].

### 3.2. Batch Adsorption Experiments

The effects of the operational parameters on the removal of CR dye, as summarised in Table 1, are illustrated in Figure 7.

#### Effect of Initial pH, Temperature and Adsorbent Dosage

As illustrated in Figure 7a, the highest percentage of removal was observed at a pH range of 3–8, after which the removal started to decrease. Mahapatra et al. [54] explained this behaviour as being a result of the interaction between hydroxide groups from the adsorbent and amine groups from CR dye.

As illustrated in Figure 7b, it was observed that the percentage removal increased with an increase in temperature and became constant after 35 °C. Therefore, 35 °C is an ideal temperature for the maximum removal of CR dye from an aqueous system.

As illustrated in Figure 7c, 1 g of the material was observed to remove 100% of CR dye. The complete removal of CR dye from an aqueous solution by PDFe/Al composite signifies the high efficiency of the material and enough availability of the adsorption sites.

### 3.3. Adsorption Kinetics

Adsorption kinetics for the adsorption of CR dye by PDFe/Al composite were studied to elucidate the varying mechanisms and rates of adsorption. The kinetic models are summarised in Table 3 with their graphical representations illustrated by Figure 8. The model parameters for the analytical form of the kinetic models in Table 3 were determined using the built-in iterative non-linear regression algorithm within the software package Graphpad Prism 9 (GraphPad Software Inc., San Diego, CA, USA).

Figure 8a shows that good fits for the data were obtained from the pseudo-first-order (PFO) model (R^2^ = 0.991); however, there is much uncertainty in the future observations relative to CR dye adsorption application, as seen in the very large 95% prediction interval due to the model’s dependence on initial adsorbate concentrations, thus limiting the application of the PFO kinetic model. In Figure 8b, the pseudo-second-order (PSO) results were obtained in the same manner as PFO; however, the PSO suffers from the same limitations in terms of application as the PFO model, due to the dependence of the model on the initial adsorbate concentration.

In Figure 8c, the two-phase adsorption (TPA) model provides an improvement on the PFO and PSO kinetic models based on two parallel adsorption processes; namely, a rapid and a slow adsorption mechanism. The results obtained show a good fit (R^2^ = 0.997) for the adsorption of CR dye. The TPA model provides insight into the mechanism of adsorption as well as the relative dominance of the adsorption kinetics on the adsorbent surface. Similar to what was proposed by Wang et al. [57], the current system involves compartmentalised adsorption with fast and slow adsorption taking place in parallel. In this system, the dominant adsorption takes place rapidly (86.3% of CR adsorbs during the fast phase) while relatively less adsorption follows slow adsorption kinetics (13.7% of CR adsorbs during the slow phase). This has significant implications for the scaling of the process, as it shows that more than 80% of the CR is removed within the first 2 min, followed by much slower adsorption. In addition, this kinetic model hints at multiple surface sites present on the surface with different adsorption characteristics [57]—this is explored further in the subsequent (Section 3.4). Figure 8d shows the Crank diffusion model, which was studied to determine the pore diffusion, based on the assumption that the adsorbent can be modelled as a porous sphere with constant effective diffusivity *(D_e_)* [55]. The results obtained show a remarkably good fit for the CR adsorption data; a comparison of the fitted effective diffusivity (*D_e_* = 3.49 × 10^−11^ m^2^·s^−1^) and the molecular diffusivity of CR (2.16 × 10^−9^ m^2^·s^−1^ [60]) showed that the system suffered from severe mass transfer limitations. Figure 8e shows the Weber–Morris intraparticle diffusion model, which was investigated to study the effect of interparticle and intraparticle diffusion on the adsorption of CR dye. In this model, the first step of adsorption is intraparticle diffusion, the second phase is intraparticle, and the last phase is adsorption onto the adsorbent [61]. The data was fitted well (R^2^ = 0.999), and the findings were good, where *D_e_*_1_ = 4.30 × 10^−12^ m^2^·s^−1^ and *D_e_*_2_ = 6.47 × 10^−15^ m^2^·s^−1^, supporting the observations from the Crank model (Figure 8d) of significant diffusion limitations in the system.

### 3.4. Adsorption Isotherms

Adsorption isotherms for the adsorption of CR dye by PDFe/Al were studied to elucidate the different mechanisms governing the adsorption process. The isotherm models are summarised in Table 4 with their graphical representations illustrated in Figure 9. The model parameters for the non-linear forms of the isotherm models in Table 4 were determined using Graphpad Prism 9 (GraphPad Software Inc., San Diego, CA, USA). The built-in non-linear regression algorithm involves an iterative process to minimise the error values between the measured data and predicted values.

Table 4 provides a summary of the adsorption isotherms from the literature, where it can be observed that the models fitted the experimental data with a coefficient of determination *R^2^* < 0.99 and a root-mean-square error RMSE >12.7 mg·g^−1^, indicating that no single isotherm model was able to conclusively predict the equilibrium data. However, sufficiently good fits were obtained to provide some insights into the mechanisms responsible for adsorption.

Figure 9a shows the Langmuir adsorption isotherm where a maximum adsorption capacity of the material (*Q_max,L_*) of 411 mg·g^−1^ was predicted. This is exceptionally high when compared to other reported studies. Of further interest is the value *R_L_* = 0.775, indicating favourable adsorption conditions—0 < *R_L_* < 1 is considered favourable for adsorption, *R_L_ >* 1 unfavourable for adsorption, *R_L_* = 1 linear adsorption, and *R_L_ =* 0 irreversible adsorption [62]. This maximum adsorption capacity is not necessarily the true maximum adsorption of the adsorbent, as the Langmuir model only fits the data with a coefficient of determination of R^2^ = 0.922.

Figure 9b shows the Freundlich adsorption isotherm, where the intensity parameter (*n_F_*) gives an indication of the favourability of the adsorption—*n**_F_* < 1 is considered unfavourable, *n_F_* = 1 designates linear adsorption, and *n_F_* > 1 shows favourable adsorption [63,64]. In this system, *n_F_* =1.99 indicates highly favourable adsorption.

Figure 9c shows the two-surface Langmuir adsorption isotherm, which assumes that the surface of the adsorbent has multiple surface types with different adsorption properties.

**Table 4 nanomaterials-12-00776-t004:** Isotherms and accompanying fitting parameters used to model the CR isotherm data.

Isotherm	Non-Linear Form *	Fitted Parameters *	R^2^/RMSE
Langmuir [55,65,66]	$Q_{e}=\frac{k_{L}Q_{max,L}C_{e}}{1+k_{L}C_{e}},$	$k_{L}=0.0290\;L\cdot mg^{-1}$ $Q_{max}=411\;mg.g^{-1}$ $R_{L}=0.775$	0.922/ 17.7 mg·g^−1^
Freundlich [66]	$Q_{e}=K_{F}C_{e}^{\frac{1}{n}}$	$K_{F}=32.9\;mg\cdot g^{-1}{\left(L\cdot mg^{-1}\right)}^{\frac{1}{n_{F}}}$ $n_{F}=1.99$	0.950/ 14.2 mg·g^−1^
Two-surface Langmuir [65]	$Q_{e}=\overset{2}{\underset{i=1}{\sum }}\frac{k_{L,i}\left(T\right)Q_{max,i}C_{e}}{1+k_{L,i}\left(T\right)C_{e}},$	$Q_{max,1\;}=33.0\;mg\cdot g^{-1},$ $k_{L,1}=6.802\;L\cdot mg^{-1}$ $Q_{max,2\;}=1270\;mg\cdot g^{-1}$ $k_{L,2}=0.004818\;L\cdot mg^{-1},$ $R_{L,1}=0.0145$ $R_{L,2}=0.95$	0.968 11.34 mg·g^−1^
Dubinin–Radushkevich [66]	$Q_{e}=Q_{max,DR}\exp\left\{-{\left[\frac{RTln\left(C_{S}/C_{e}\right)}{E_{DR}}\right]}^{2}\right\}$	$E_{DR}=15.0\;kJ\cdot mol^{-1}$ $Q_{max,\mathrm{DR}}=658\;mg\cdot g^{-1}$	0.936/ 16.1 mg·g^−1^
Dubinin–Astokov [66]	$Q_{e}=Q_{max,DA}\exp\left\{-{\left[\frac{RTln\left(C_{S}/C_{e}\right)}{E_{DA}}\right]}^{n_{DA}}\right\}$	$E_{DA}=14.7\;kJ\cdot mol^{-1}$ $Q_{max,DA\;}=658\;mg\cdot g^{-1}$ $n_{DA}=1.85$	0.937/ 15.9 mg·g^−1^
Sips [67]	$Q_{e}=\frac{K_{S}Q_{max,S}C_{e}^{\frac{1}{n_{S}}}}{1+K_{S}C_{e}^{\frac{1}{n_{S}}}}$	$Q_{max,S}=1350\;mg\cdot g^{-1}$ $K_{S}=0.0261\;g\cdot g^{-1}{\left(L\cdot mg^{-1}\right)}^{\frac{1}{n_{S}}}$ $n_{S}=1.93$	0.947/14.6 mg·g^−1^

* The definitions of the isotherm parameters: Langmuir isotherm: *k_L_*—the Langmuir equilibrium constant (L·mg^−1^); *Q_max,L_*—the Langmuir maximum adsorption capacity (mg·g^−1^); *R_L_*—the Langmuir separation factor (dimensionless). Freundlich isotherm: *K_F_*—the Freundlich intensity parameter ((mg·g^−1^)(L·mg^−1^)^1/nF^); *n_F_*—Freundlich isotherm exponent (dimensionless). Two-surface Langmuir isotherm: *k_L,i_*—the Langmuir equilibrium constant for surface *i* (L·mg^−1^); *Q_max,i_*—the Langmuir maximum adsorption capacity for surface *i* (mg·g^−1^); *R_L,i_*—the Langmuir separation factor for surface *i* (dimensionless). Sips isotherm: *K_S_*—the Sips constant ((mg·g^−1^)(L.mg^−1^)^1/nS^); *Q_max,S_*—the Sips maximum adsorption capacity (mg·g^−1^); *n_F_*—Sips isotherm exponent (dimensionless). Dubinin–Radushkevich (DR) isotherm: *Q_max,DR_*—the DR maximum adsorption capacity (mg·g^−1^); *R*—the ideal gas constant (8.314 J·mol^−1^·K^−1^); *T*—temperature in the system (K); *C_S_*—saturation concentration of the adsorbate (33 g·L^−1^ [68]); *E_DR_*—energy required to adsorb an adsorbate from infinity to the adsorbent surface (kJ·mol^−1^). Dubinin–Astokov (DA) isotherm: *Q_max,DA_*—the DA maximum adsorption capacity (mg·g^−1^); *T*—temperature in the system (K); *C_S_*—saturation concentration of the adsorbate (33 g·L^−1^ [68]); *E_DA_*—energy required to adsorb an adsorbate from infinity to the adsorbent surface (kJ·mol^−1^); *n_DA_*—DA isotherm exponent (dimensionless).

The results from this model were markedly better than that for the single surface Langmuir model with *Q_max,_*_1_ = 33.0 mg·g^−1^ and *Q_max,_*_2_ = 1270 mg·g^−1^ for two surfaces, giving a total maximum adsorption of 1303 mg·g^−1^—significantly greater than that previously reported for CR adsorption. However, since the total maximum adsorption capacity is a gross extrapolation of the experimental data, the more conservative Langmuir adsorption capacity (*Q_max,L_*) was adopted as the official maximum adsorption capacity for the study. What is of most interest is that the two-surface Langmuir model proposes that the adsorbent surface consists of two separate adsorption surfaces with different energies and sorption properties on which parallel adsorption takes place [65] (likely related to TPA kinetics—Table 3). The first surface exhibited irreversible adsorption with *Q_max,_*_1_ = 33.0 mg·g^−1^, while the second adsorption surface exhibited reversible adsorption with a remarkably large *Q_max,_*_2_ = 1270 mg·g^−1^. This is supported by the observations that the *R_L,_*_1_ = 0.0145 indicates nearly irreversible adsorption, while *R_L,_*_2_ = 0.95 indicates highly favourable adsorption conditions [62].

Figure 9d shows the Dubinin–Radushkevich (DR) isotherm, which predicts dilute liquid adsorption behaviour [69]. This isotherm was subsequently generalised to the Dubinin–Astokov (DA) isotherm (Figure 9e), which accounts for non-ideal adsorption behaviour [70]. Results from the DR and DA non-linear fits (Table 4) illustrate comparable adsorption capacities to Langmuir and two-surface Langmuir models (*Q_max,DR_* = 658 mg·g^−1^ and *Q_max,DA_* = 658 mg·g^−1^). The adsorption energies for *D_R_* and *D_A_* isotherms were determined and found to be *E_DR_* = 15.0 kJ·mol^−1^ and *E_DA_* = 14.7 kJ·mol^−1^. The values for the adsorption energies indicate a diffusion-controlled adsorption mechanism—energies >16 kJ·mol^−1^ designate chemisorption, energies <8 kJ·mol^−1^ correspond to physisorption, and energies between 8 and 16 kJ·mol^−1^ indicate mass transfer limited diffusion [71]

Figure 9f shows the Sips adsorption isotherm, where *Q_max,S_* = 1350 mg·g^−1^, *K_S_* = 204.15 and *n_S_* = 1.93 for the Sips isotherm. According to Keren et al. [72], the Sips adsorption isotherm is a flexible isotherm model which corresponds to different isotherm models depending on the value for *n_S_*. The Sips isotherm reduces to the Langmuir isotherm for *n*_S_ = 1, while a value for *n*_S_ > 1 results in a sorbate–sorption site interaction characterised by a Gaussian distribution (known as the Langmuir–Freundlich model). A value for *n*_S_ < 1 indicates a cooperative reaction between sorption sites and *n* sorbate molecules; this type of adsorption is characterised by sigmoidal adsorption isotherm data. The Sips isotherm model further reduces to the Freundlich model for *n_S_* > 1 and at low adsorbate concentrations (when $K_{S}C^{1/ns}$ ≪ 1). The implication for the current study is that the system corresponds to the Langmuir–Freundlich model which shows a distributive adsorbate–sorption site interaction that deviates significantly from the Langmuir isotherm model but corresponds well with the Freundlich model at low adsorbate loadings. The value for *n_S_* = 1.93 shows that favourable adsorption conditions are present (similar to the Freundlich isotherm). The Sips isotherm model links strongly to the two-surface Langmuir model which predicts a heterogeneous surface with two main surface sites showing differing adsorption characteristics.

The values for the *Q_max_* for both the two-surface Langmuir and Sips models are >1300 mg·g^−1^, which is remarkably high for CR adsorption. However, as indicated before due to the uncertainty inherent in such significant extrapolation, the *Q_max,L_* was adopted as the official maximum adsorption capacity of the study.

### 3.5. Comparison of PDFe/Al with Other Adsorbents

Table 5 summarises prominent CR adsorption studies using Al- and Fe-based adsorbents from the literature, thereby providing a comparison of the results for the current study. From Table 5, it can be seen that the current study compares well with other Fe- and Al-based adsorbents for CR adsorption in terms of adsorption capacity (*Q_max,L_*) and the favourability of adsorption (*n_F_* > 1). This is especially pertinent considering that the only materials to outcompete the current materials in terms of *Q_max,L_* were synthesised using laboratory-grade chemicals, whereas the PDFe/Al was synthesised using *bona fide* AMD. In addition, the materials showing higher adsorption favourability (higher *n_F_*) displayed significantly lower maximum adsorption capacities, indicating that PDFe/Al represented a highly desirable combination of high adsorption capacity and favourability towards CR adsorption.

### 3.6. Regeneration Study

As illustrated in Figure 10, a regeneration study was performed to regenerate the material after CR dye adsorption.

The material showed the potential to be reused more than four times for CR dye adsorption. It should be noted that even during the sixth regeneration cycle the % desorption ≈ 95%, indicating a significantly stable and regenerable adsorbent for industrial application.

### 3.7. Adsorption Mechanism

The proposed adsorption mechanism for CR adsorption is summarised in Figure 11.

The adsorption of CR dye by PDFe/Al appears to be predominantly diffusion-controlled, with the greatest predicted effective diffusivity of 3.49 × 10^−11^ m^2^·s^−1^ being two orders of magnitude smaller than the molecular diffusivity of CR of 2.16 × 10^−9^ m^2^·s^−1^ [60]. This is further supported by the predicted Dubinin–Radushkevich and Dubinin–Astokov energies of adsorption of *E_DR_* = 15.0 kJ·mol^−1^ and *E_DA_* = 14.7 kJ·mol^−1^, both indicating a diffusion-controlled adsorption mechanism [71]. The diffusion-controlled system is likely a reason for the difficulty in conclusively determining if physisorption or chemisorption is dominant in the system (relatively poor fits for both PFO and PSO kinetics models). The two-surface Langmuir model indicated that the adsorbent consisted of two distinct adsorption surfaces, one irreversible with an adsorption capacity of 33 mg·g^−1^ and one reversible with and adsorption capacity of 1270 mg·g^−1^. This multi-surface characteristic is supported by two-phase adsorption kinetics, showing a fast and slow adsorption process taking place in parallel. Finally, it was observed that the adsorption of CR on the adsorbent surface induced significant internal strain within the adsorbent, which resulted in deformation, cracking, and subsequent increased adsorption area and volume within the adsorbent. This increased area, coupled with electrostatic interactions and additional covalent/donor–acceptor interactions between the surface of the adsorbent and CR dye ions, could explain the high adsorption of CR dye ions by PDFe/Al [80].

## 4. Practical Implications of This Study

This study has demonstrated multiple technical feasibilities in the use of the synthesised nanocomposite for the removal of CR dye. Henceforth, upscaling this technology to a viable industrial scale could be possible, specifically considering the amount of mine water being generated in South Africa. Furthermore, findings from our laboratory assays have confirmed the efficacy of the synthesised PDFe/Al; the next stage should be to pilot the application for the removal of CR using real industrial effluents.

Future work and Recommendations:Column tests will be carried out to assess the developed materials’ ability to remove CR continuously.A techno-economic analysis of the suggested material should be performed to determine the technology’s economic viability.The doping of the material should be considered to boost its adsorption capacity, hence increasing the proposed materials’ adsorption efficiency.A life-cycle analysis of the proposed system will be used to determine the environmental sustainability of the technology.

## 5. Conclusions

This double-edged study presents the first known investigation into the adsorption of CR by PDFe/Al synthesised from *bona fide* AMD. The study successfully demonstrated the highly effective removal of CR from aqueous solution by PDFe/Al. The optimised conditions for CR dye adsorption were found to be: 100 mg·L^−1^ initial CR dye concentration, 1 g of PDFe/Al in 500 mL adsorbate solution, pH 3–8, 20 min agitation, and a temperature of 35 °C. The synthesised PDFe/Al had a Langmuir adsorption capacity of 411 mg·g^−1^ CR dye with >99% removal of CR—one of the highest for CR adsorption. The adsorption mechanism was found to be significantly diffusion-limited, with a dual-site adsorption surface responsible for adsorption. It was further found that CR adsorption induced strain and deformation within the PDFe/Al, resulting in internal crack formation and a consequent increase in pore surface area (37.58 m^2^.g^−1^ before adsorption vs 134.46 m^2^.g^−1^ after adsorption) and pore volume (0.0621 cm^3^. g^−1^ before adsorption vs 0.1137 cm^3^. g^−1^ after adsorption). This remarkable increase in the surface area could provide an explanation of the notable adsorption capacity measured. Finally, it was determined that the PDFe/Al could be regenerated and reused more than four times with minimal loss in adsorption efficiency. The current study successfully valorised and beneficiated AMD and provides an avenue to foster and grow the circular economy in wastewater treatment.

## Figures and Tables

**Figure 1 nanomaterials-12-00776-f001:**
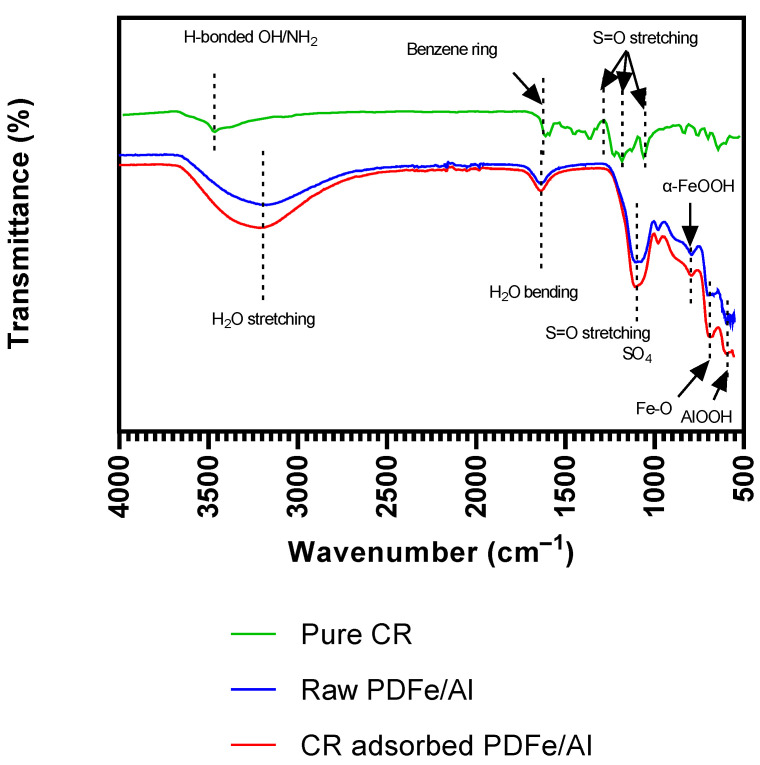
Functional groups of the synthesised PDFe/Al before and after CR dye adsorption.

**Figure 2 nanomaterials-12-00776-f002:**
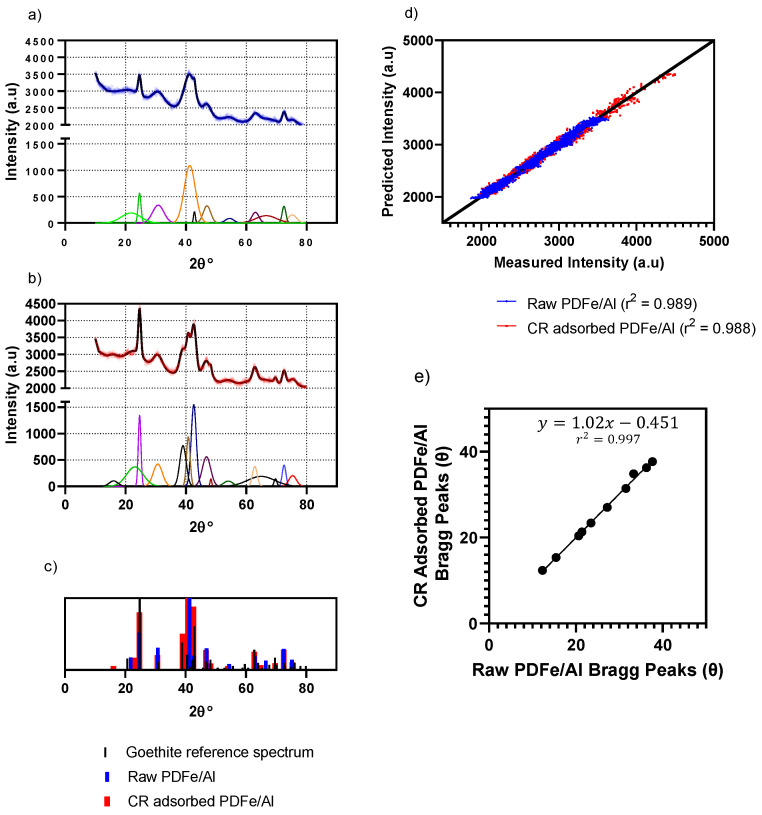
XRD analysis of the PDFe/Al composite (**a**) before and after (**b**) CR dye adsorption. The coloured peaks represent the deconvoluted peaks extracted from the XRD data (blue data in (**a**) and red data in (**b**)) with offset subtracted. (**c**) The relative peaks from (**a**,**b**) against the reference spectrum for goethite [22], indicating the positive characterisation of goethite in the sample. (**d**) The parity plot of the experimental data obtained from the XRD diffractometer and the fitted Gaussian peaks shown in (**a**,**b**), demonstrating the quality of the fitted curves. (**e**) Plot of the Bragg peak positions for (**a**) against the corresponding peaks in (**b**). The plot shows that a constant shift of −0.451° was measured after CR adsorption as compared to the raw PDFe/Al.

**Figure 3 nanomaterials-12-00776-f003:**
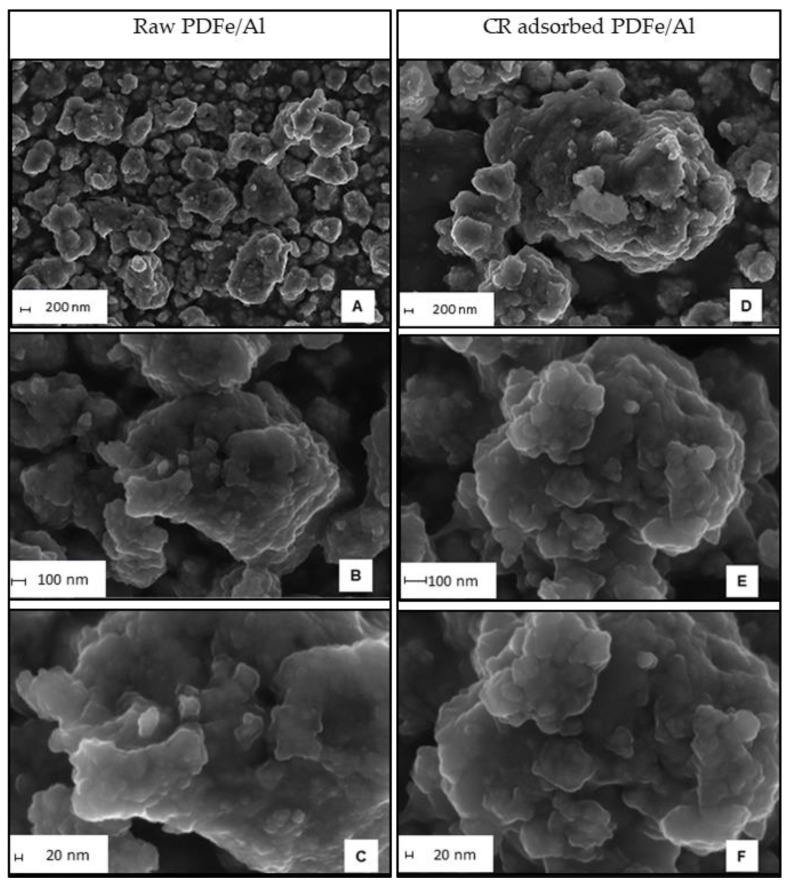
Morphology of the synthesised PDFe/Al before adsorption (**A**–**C**) and after CR dye (**D**–**F**) adsorption. The scale bars are shown in the bottom left of the figures.

**Figure 4 nanomaterials-12-00776-f004:**
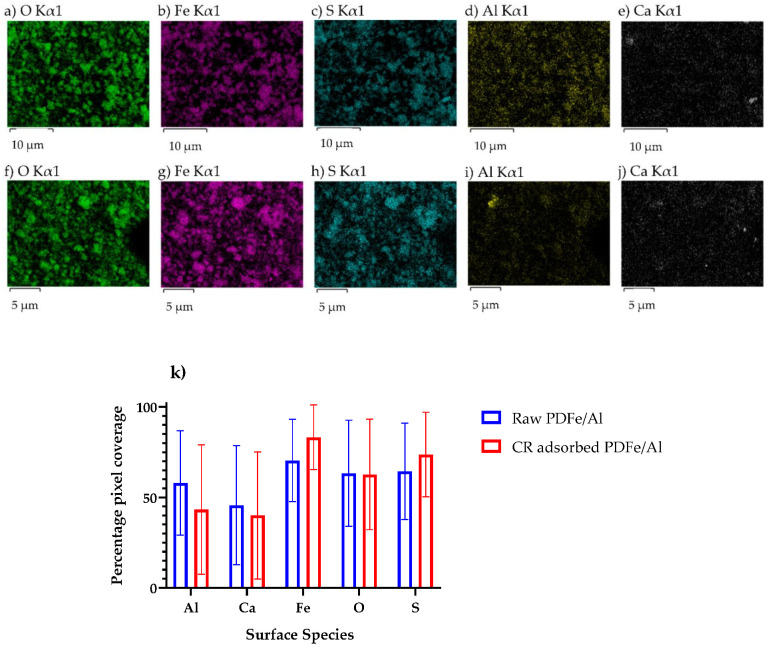
(**a**–**e**) Mapping of the elemental composition of the synthesised PDFe/Al before adsorption. (**f**–**j**) Mapping of the elemental composition of the synthesised PDFe/Al after CR adsorption. (**k**) The percentage pixel coverage of respective elements from EDX mapping. Error bars show standard deviations.

**Figure 5 nanomaterials-12-00776-f005:**
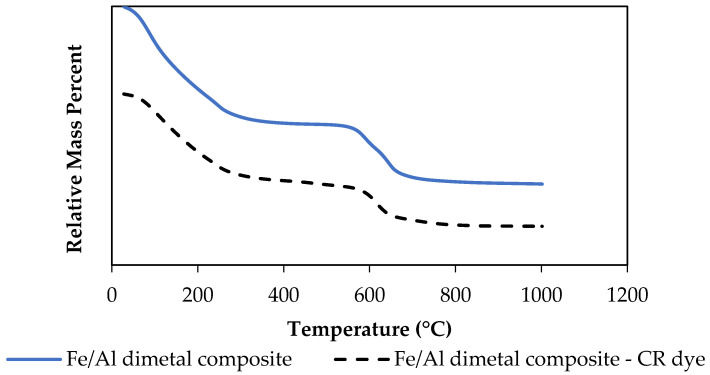
Thermal stability of the synthesised PDFe/Al before and after CR dye adsorption.

**Figure 6 nanomaterials-12-00776-f006:**
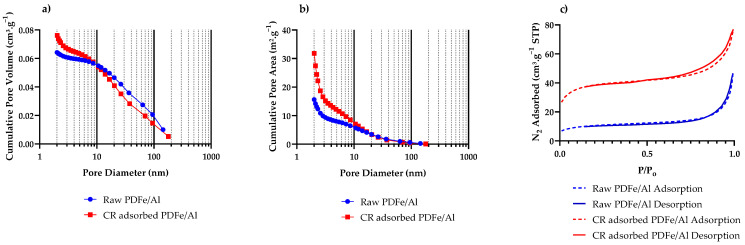
BET surface area and porosity analysis of PDFe/Al before and after CR dye adsorption. (**a**) The pore volume distribution, (**b**) the pore area distribution, and (**c**) the N_2_ adsorption hysteresis loops for raw and CR-adsorbed PDFe/Al.

**Figure 7 nanomaterials-12-00776-f007:**
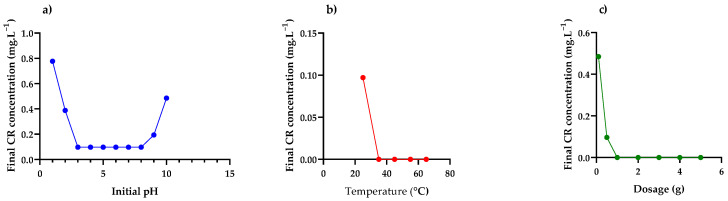
Effects of (**a**) Initial system pH, (**b**) Temperature, (**c**) Adsorbent dosage on the adsorption of CR dye.

**Figure 8 nanomaterials-12-00776-f008:**
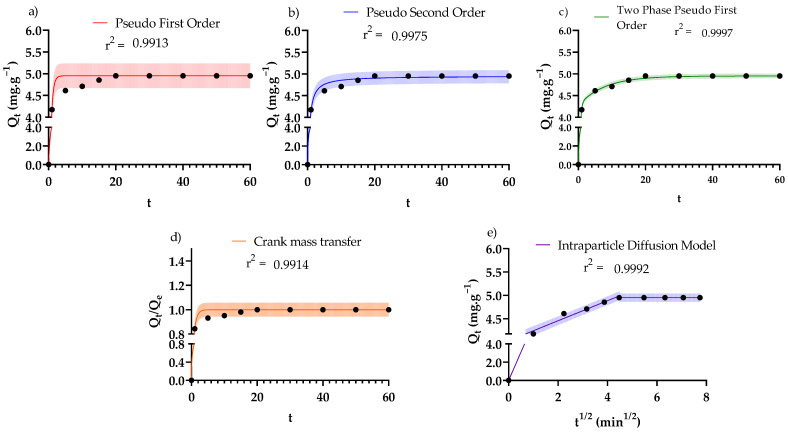
Different kinetic models fitted to the kinetic data for the adsorption of CR dye to the adsorbent. The models were: (**a**) pseudo-first-order, (**b**) pseudo-second-order, (**c**) two-phase pseudo-first-order, (**d**) Crank mass transfer model, and (**e**) intraparticle diffusion model. The shaded areas represent the 95% prediction intervals for the respective model fits. The optimised model parameters are reported in Table 3.

**Figure 9 nanomaterials-12-00776-f009:**
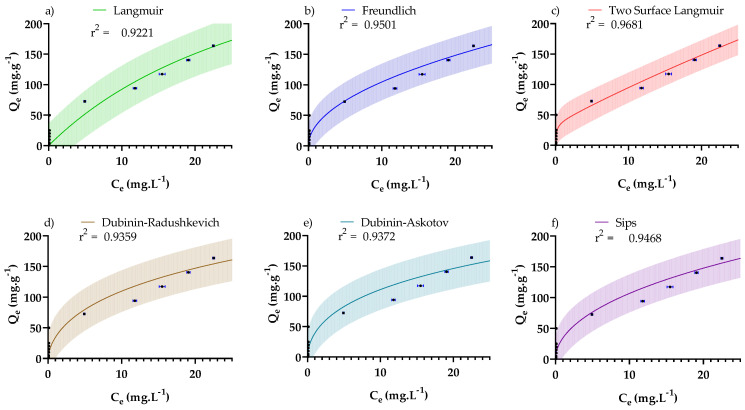
The non-linear fits of the isotherm models from (**a**) Langmuir, (**b**) Freundlich, (**c**) Two Surface Langmuir, (**d**) Dubinin–Radushkevich, (**e**) Dubinin–Askotov, and (**f**) Sips. The shaded areas indicate the 95% prediction intervals.

**Figure 10 nanomaterials-12-00776-f010:**
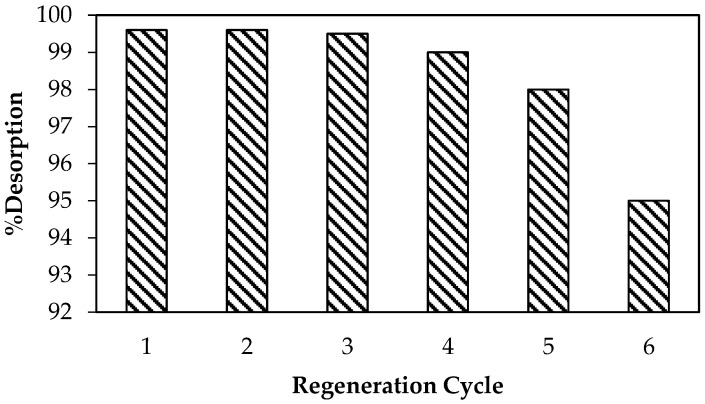
Recovery efficiency of PDFe/Al after CR dye adsorption.

**Figure 11 nanomaterials-12-00776-f011:**
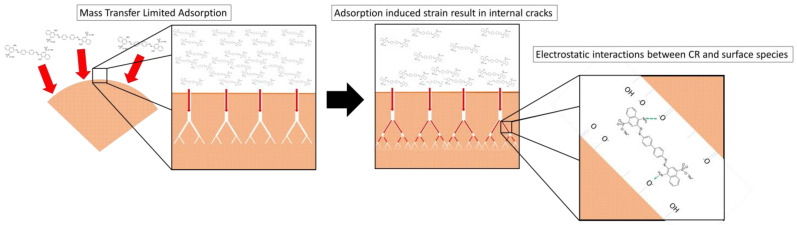
Proposed adsorption mechanism.

**Table 1 nanomaterials-12-00776-t001:** Studied parameters for CR dye adsorption using PDFe/Al.

Experiment No	Initial Concentration (mg·L^−1^)	Initial pH	Adsorbent Dosage * (g)	Agitation Time (min)	Temperature (°C)
1	1; 5; 10; 20; 30; 40; 50; 100; 150; 200	6–7	1	30	25
2	10	2, 3, 4, 5, 6, 7, 8, 9, 10 (±0.2)	1	30	25
3	10	6–7	0.1; 0.5; 1; 2; 3; 4; 5 (±0.0005)	30	25
4	10	6–7	1	1; 5; 10; 15; 20; 25; 30; 40; 50; 60	25
5	10	6–7	1	30	25; 35; 45; 55; 65

* Dispersed in a 500 mL batch reactor.

**Table 2 nanomaterials-12-00776-t002:** Surface area and porosity properties of PDFe/Al before and after CR dye adsorption.

Parameter	Raw PDFe/Al	CR-PDFe/Al
BET Surface Area (m^2^·g^−1^)	37.58 ± 0.37	134.46 ± 2.06
Micropore Area (from t-Plot) (m^2^·g^−1^)	13.67	51.85
Pore volume (<179 nm) (cm^3^·g^−1^)	0.0621	0.1137
Micropore Volume (cm^3^·g^−1^)	0.0031	0.0155
BJH average pore diameter (nm)	16.47	9.58

**Table 3 nanomaterials-12-00776-t003:** Kinetic models for the adsorption of CR dye by PDFe/Al composite.

Kinetic Law	Differential Form *	Analytical Form *	Fitted Parameters *	R^2^/RMSE
Pseudo-first-order [55]	$\frac{dQ_{t}}{dt}=k_{1}\left(Q_{e}-Q_{t}\right)$	$Q_{t}=Q_{e}\left(1-e^{-k_{1}t}\right)$	*k*_1_ = 1.85·min^−1^	0.991/ 0.138 mg·g^−1^
Pseudo-second-order [55]	$\frac{dQ_{t}}{dt}=k_{1}{\left(Q_{e}-Q_{t}\right)}^{2}$	$Q_{t}=\frac{\left(k_{2}Q_{e}^{2}\right)t}{1+k_{2}Q_{e}t}$	*k*_2_ = 1.00 g·mg^−1^·min^−1^	0.998/ 0.0734 mg·g^−1^
Two-phase adsorption [56,57,58]	$\frac{dQ_{t,slow}}{dt}=k_{slow}\left(\left(1-\phi \right)Q_{e}-Q_{t,sloe}\right)$ $\frac{dQ_{t,fast}}{dt}=k_{fast}\left(\phi Q_{e}-Q_{t,fast}\right)$ $\frac{dQ_{t}}{dt}=\frac{dQ_{t,slow}}{dt}+\frac{dQ_{t,fast}}{dt}$	$Q_{t}=Q_{e}\left(\left(1-\phi \right)\left(1-e^{-k_{fast}t}\right)+\phi \left(1-e^{-k_{slow}t}\right)\right)$	*k_fast_* = 3.18 min^−1^ *k_slow_* = 0.128 min^−1^ *ϕ* = 0.863	1.000/ 0.0263 mg·g^−1^
Crank internal mass transfer model [55]	$\frac{\partial Q_{t}}{\partial t}=\frac{D_{e}}{r^{2}}\frac{\partial }{\partial r}\left(\frac{r^{2}\partial Q_{t}}{\partial r}\right)$	$\frac{Q_{t}}{Q_{e}}=\frac{6}{\pi^{2}}\overset{\infty }{\underset{n=1}{\sum }}\frac{1}{n^{2}}\exp\left(-\frac{D_{e}}{r^{2}}n^{2}\pi^{2}t\right)$	*D_e_* = 3.49 × 10^−11^ m^2^·s^−1^	0.991/0.0277 mg·g^−1^
Weber and Morris [55,59]		$Q_{t}=k_{WM,i}t^{\frac{1}{2}}+C,$ $k_{WM,i}=\frac{D_{e,i}}{r^{2}}$	*D_e_*_1_ = 3.49 × 10^−11^ m^2^·s^−1^ *D_e_*_2_ = 4.19 × 10^−14^ m^2^·s^−1^ *D_e_*_3_ = 0 m^2^·s^−1^	0.999/ 0.0421 mg·g^−1^

* The definitions of the kinetic model parameters: General parameters: *Q_t_*—amount of dye adsorbed per unit of adsorbent at time t (mg·g^−1^); *Q_e_*—equilibrium adsorption capacity of adsorbent (mg·g^−1^); *r* = the average radius of the adsorbent particles (a conservative estimate of 16 μm was used as all particle diameters <32 μm). Pseudo-first-order (PFO) kinetics: *k*_1_—the PFO rate constant (min^−1^). Pseudo-second-order (PSO) kinetics: *k*_2_—the PSO rate constant (g·mg^−1^·min^−1^). Two-phase adsorption (TPA) kinetics: *k_fast_*—the rate constant for the fast TPA adsorption (min^−1^); *k_slow_*—the rate constant for the slow TPA adsorption (min^−1^); *ϕ*—the fraction of adsorption taking place during the fast adsorption step (dimensionless). Crank internal mass transfer kinetic model: *D_e_*—the effective diffusivity of the adsorbate in the system (m^2^·s^−1^). Weber and Morris kinetic model: *K_WM,i_*—the Weber-Morris intra-particle diffusion rate constant for adsorption phase *i* (mg·g^−1^); *D_e,i_*—the effective diffusivity of the adsorbate during adsorption phase *i* (m^2^·s^−1^).

**Table 5 nanomaterials-12-00776-t005:** Comparison of PDFe/Al with other Al- and Fe-based adsorbents for CR adsorption in order of decreasing *Q_max,L_* values.

Adsorbent	*Q_max,L_* *	*n_F_* **	Reference
BTCS functionalised Fe_3_O_4_ nanoparticles	630 mg·g^−1^	1.9596	[73]
γ-Fe_2_O_3_–γ-Al_2_O_3_	416.7 mg·g^−1^	1.13	[54]
PDFe/Al	411 mg·g^−1^	1.99	This study
Iron doped PVA-chitosan	315 mg·g^−1^	6.34	[74]
Fe_x_Co_3−x_O_4_	160.3 mg·g^−1^	1.44	[75]
Calcium Alginate Beads—nano-goethite	181.1 ± 2.32 mg·g^−1^	2.431 ± 0.343	[76]
Fe_2_O_3_–Al_2_O_3_	126.58 mg·g^−1^	1.85	[54]
Graphine Oxide-CuFe_2_O_4_	114.21 mg·g^−1^	1.223	[77]
Alumina-Zirconia	41.07 mg·g^−1^	2.177	[78]
Nano bio-clay composite (Kaolinite/Ulva Lactuca)	23.7529 mg·g^−1^	1.68	[79]
α-Fe_2_O_3_–α-Al_2_O_3_	1.422 mg·g^−1^	−0.665	[54]

* *Q_max,L_*—the Langmuir maximum adsorption capacity (mg·g^−1^); ** *n_F_*—the Freundlich isotherm exponent (dimensionless).

## Data Availability

The data presented in this study are openly available in the University of Pretoria Research Data Repository at doi:10.25403/UPresearchdata.19228875.

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
