# Peer review of "Effective Adsorption of Congo Red from Aqueous Solution Using Fe/Al Di-Metal Nanostructured Composite Synthesised from Fe(III) and Al(III) Recovered from Real Acid Mine Drainage"

_nanomaterials, 2022, doi:10.3390/nano12050776_

Round 1

Reviewer 1 Report

Muedi et al. have aimed to recover and synthesize polycationic Fe/Al Di-metal nanostructured composite from acid mine drainage and apply as an adsorption for Congo red dye. The work has been carried out systematically and should be of interest in this field. However, there are several issues the authors need to address before this paper could accepted for publication.

  1. The authors should include information on global statistics on annual generation of AMD.
  2. How do the authors claim the synthesized nanocomposite to be of nanosized? Did you measure the particle size?
  3. P3, L123-132 – Different initial concentration, initial pH, adsorbent dosage, agitation time and temperature should also be provided in the form of range in the text within parenthesis as 1-200 mg/L, 2-10, 0.1-5 g, 1-60 min and 25-65°C, respectively. By the way, is the adsorbent dosage given as g or g/L in Table 1?
  4. P3, L119 – “For quality control and -assurance” should be corrected as “For quality control and quality assurance”.
  5. Figure 4 - The labels should be bigger in size to enable visibility and clarity.
  6. Table 2 – the number of digits after the decimal point should be controlled the values 37.5841, 134.4634, 13.6686, 51.8486, 0.062053, 0.113727, 0.003066, 0.015451, 16.4650 and 9.5797 should corrected as 37.58, 134.46, 13.67, 51.85, 0.0621, 0.1137, 0.0031, 0.0155, 16.47 and 9.58, respectively.
  7. Sections 3.2.1, 3.2.2, 3.2.3 – these sections should be combined to be under one common heading as “Effect of initial pH, temperature and adsorbent dosage”.
  8. Tables 3-5 – as the terms involved in various kinetic and isotherm equations are not described elsewhere in the text, these terms should be described in the foot note of respective tables.
  9. P12, L340 – what is i.t.o application? Please expand it.
  10. Table 4 – the Sips isotherms is balanced model intermediate between Freundlich and Langmuir models. Therefore, the exponent value of n being lesser or greater than 1 has got some significant information to be discussed. More specifically, the n value of 1.93 which is greater than 1 should be discussed, as the n<1 means the isotherm approaching Freundlich and n=1 means the Langmuir isotherm approaching Langmuir and likewise the authors should discuss what it means if n>1. This information will aid to refine/add-on the adsorption mechanism described in this study. The authors may refer to this paper which contains some brief discussion on this: Chemosphere 2015, 138, 462-468.
  11. Some relevant references should be cited such as Bioresource Technology 2011, 102, 8868-8876 and Journal of Hazardous Materials 2021, 415, 125701.
  12. P14, L388 – “different energetic and sorption properties” should be corrected as “different energies and sorption properties”.
  13. Notations of units are used interchangingly in text, tables and figures. For instance, mg/L and mg.L-1. Also, hr/hours, minutes and sec/seconds should be abbreviated as h, min and s, respectively.
  14. Conclusions – it should contain only a summary of the key findings presented in this study and therefore the first paragraph should be condensed and combined with the second one.
  15. P16, L467 – “duel” should be corrected as “dual”.

Author Response

Please see the attached document for details on the manuscript revisions and responses to the reviewer comments.

Reviewer 2 Report

  1. How authors calculated strain and dislocations?
  2. The authors need to improve the introduction section.  The chemistry between Al and Fe metal is missing in the introduction section for the CR dye adsorption. 
  3. Authors should discuss the novelty of the work.
  4. The introduction section lacks the discussion of known other materials/metals used for the CR dye adsorption. How Fe/Al Di-metal 2
    nanostructured composites are efficient compared to the reported well-known materials. 
  5. Authors should explicitly specify the novelty of their work. What progress against the most recent state-of-the-art similar studies was made in this study? Mention this in the revised manuscript. For instance, the last paragraph or closing lines of the introduction section always highlight the novelty aspects of the study with the clear aim of the study and the importance/significance of the study findings.
  6. Under the section on results and discussion, it is recommended to discuss and explain what should be the appropriate policies based on the findings of this study. Also, the results should be further elaborated to show how they could be used for real applications.
  7. It is strongly recommended to add a subsection, 'practical implications of this study,' outlining the challenges in the current research, future work, and recommendations, before the conclusion. Currently, the conclusions contain both the concluding marks and recommendations. The future work-related points can be grouped under the subsection mentioned above.
  8. There are such language barriers and a large number of mistyping should be checked and corrected.
  9. Check the symbols of the Figures and correct them. Like Fig. 5. Additionally, modify figures. 
  10. The mechanism is discussed, but its pictorial presentation is missing
  11. The scale of FESEM images is not clear. 

Author Response

(The authors gave the same response as above.)

Reviewer 3 Report

The authors reported the development of polycationic Fe/Al di-metal nanostructured composite (PDFe/Al) and its application on the removal of dye from waste water. The characterization aspect of the prepared composites was sufficient, and the practical aspect (dye removal) was reasonably and sufficiently studied. This manuscript needs major revision and I have listed these issues and recommendations in chronological order. Following is a summary of the major corrections and revisions:

  1. The authors may need to briefly address the difference(s) between the current manuscript and other similar published review articles in the Introduction section.
  2. In general, the references in the introduction are poorly chosen, when compared to the sentences they serve as confirmation for.
  3. Please explain why you have selected Congo red, instead of other dye.
  4. Please do not use linearization of the equations. Nowadays, most computer programs can perform non-linear regression and should be used in preference of linearization to determine adsorption parameters. Comment please.
  5. Please provide the XRD, FTIR and SEM, results of the adsorbent after the five times of regeneration.
  6. Did the authors check any interferences in spectrophotometer measurements of dye from the matrix of the synthesized materials? Please comment.
  7. Can this work be feasible to be done in industrial scale, and can it be scaled up? Can the same experiments be done using continuous adsorption column?
  8. Conclusions: Conclusions need to be improved by specifying the discussed important points within this work. In the conclusions, the authors should also provide an outlook of the challenges and potential future directions.

Author Response

(The authors gave the same response as above.)

Round 2

Reviewer 1 Report

The authors have satisfactorily addressed all the comments raised by reviewers and therefore I recommend acceptance of this article for publication in Nanomaterials.

Reviewer 3 Report

Authors have revised the manuscript according the recommendations, and answered the questioned points. Now it looks suitable for publication in Nanomaterials.